# Readable High-Speed Racetrack Memory Based on an Antiferromagnetically Coupled Soft/Hard Magnetic Bilayer

**DOI:** 10.3390/nano9111538

**Published:** 2019-10-30

**Authors:** Ziyang Yu, Chenhuinan Wei, Fan Yi, Rui Xiong

**Affiliations:** 1Key Laboratory of Artificial Micro- and Nano-structures of Ministry of Education, School of Physics and Technology, Wuhan University, Wuhan 430072, China; tommyu91@163.com (Z.Y.); nancychnwei@163.com (C.W.); xiongrui@whu.edu.cn (R.X.); 2School of Electronics and Information Engineering, Wuhan Donghu University, Wuhan 430212, China

**Keywords:** readable, high-speed, antiferromagnetically coupled, soft/hard magnetic bilayer

## Abstract

The current-induced domain wall (DW) motion in a racetrack memory with a synthetic antiferromagnets (SAFs) structure has attracted attention because of the ultrahigh velocity of the DW. However, since there is little stray field due to the zero net magnetization in a pair of antiferromagnetically (AFM) coupled domains, how to read the information stored in the pair of domains is still challenging. In the present work, we propose a readable SAF racetrack memory composed of two ferromagnetic (FM) layers with distinct uniaxial-anisotropy constants. As a result, a region of staggered domains formed between two neighboring DWs in the two layers. In this region, there is a parallel alignment of the moments in the two FM layers. This parallel magnetization is readable and can be exploited to label the structure of the nearby AFM-coupled domains for the racetrack with DWs moving in a fixed direction. This function can be realized by connecting a Schmitt Trigger to a sensor for reading. The stability and the length of the staggered region can be well-tuned by changing the magnetic parameters, such as the interlayer exchange coupling constants, the Dzyaloshinskii–Moriya interaction (DMI) constants, and the uniaxial-anisotropy constants of the two FM layers, in a range that is experimentally achievable.

## 1. Introduction

The racetrack memory is a novel magnetic memory device in which the information is labeled by the orientation of the magnetization in a magnetic domain in a nanowire, and the reading is based on the motion of magnetic domain walls (DWs) driven by a nanosecond current pulse [1,2]. Compared to other non-volatile memory devices, the racetrack memory may promise a higher storage density, a larger reading speed, and a lower dissipation.

In the early stage, the racetrack memory was made of a ferromagnetic (FM) nanowire, and the current-induced motion of DWs is attributed to the exchange of angular momentum between the spin of the moving electron and the magnetic moment in the DW (the so-called spin-transfer torque (STT) effect) [3,4,5,6,7,8]. In recent years, a new type of racetrack memory consisting of heavy metal (HM)/FM bilayer nanowires was proposed [9,10,11,12,13,14,15,16,17]. In the HM/FM bilayer track, the FM layer exhibits a perpendicular magnetic anisotropy (PMA), and the DW structure in the FM layer shows chirality due to the Dzyaloshinskii–Moriya interaction (DMI) at the HM/FM interface. This sort of DW can be driven by a current in the HM layer, which originates from the so-called spin-orbit torque (SOT) effect. Due to the combined action of DMI and SOT, the DW in the HM/FM bilayer track can be driven more efficiently than that driven by STT in the traditional FM nanowire [2].

Very recently, the HM/FM racetrack was improved by replacing the single FM layer by two FM layers with an interlayer AFM coupling (a synthetic antiferromagnetic (SAF) bilayer track). Besides SOT and DMI, another exchange torque drives the DWs to move at very high speed (close to 1000 m/s) in the SAF racetrack [2,18]. The SAF bilayer track is a promising route to develop the racetrack memory with ultra-high reading speed. However, how to read the information stored in the SAF racetrack is still challenging.

In a conventional hard disc drive (HDD) or a racetrack memory made of an FM nanowire, the information is read by measuring the stray field of the domain by a nearby reading head. In the SAF multilayer, there are two sorts of AFM structures (+M_S_ (the saturation magnetization) for the upper layer and −M_S_ for the lower layer or vice versa) that can be encoded as the digital signals 1 and 0. However, the net magnetization of the memory unit (the AFM-coupled domains) is zero, which offers no stray field for the nearby reading head. Some other techniques have also been put forward to read the information stored in a spintronic device with a traditional inter-atomic AFM exchange coupling [19,20,21,22,23], yet they seem not quite compatible with the SAF racetrack memory device. Now, Parkin’s group described a kind of drag that involves the exchange of angular momentum between two current-driven magnetic domain walls in SAF structure and the motion of DWs is determined by the strength of the drag [24]. As an alternative route, a reading head made of a magnetic tunnel junction (MTJ) can be in contact with the SAF multilayer by exploiting the upper FM layer as the free layer of the MTJ, and the digital information corresponds to the orientation of the moments in the upper FM layer. This is reasonable in principle but is factually rather challenging in experiment. In general, the uppermost layer of a SAF racetrack is not the FM layer but is a thick capping layer to avoid the oxidation. Exploiting the upper FM layer as the free layer of the MTJ is not quite experimentally compatible with the fabrication of a capping layer above the upper FM layer in SAF structure. To satisfy both requirements one needs to exploit very complicated lithography and etch technologies. 

In the present work, we report our numerical work concerning the SOT-induced motion of the DWs in a special SAF racetrack composed of two FM layers with distinct uniaxial magnetic anisotropy constants. Between two neighboring DWs in the two FM layers, a staggered domain region with the parallel alignment of the magnetic moments was generated after the DW motion was driven. The direction of the magnetization in the staggered domain region can label the digital information stored in its nearby AFM-coupled domain pair using a conventional reading method for an FM racetrack memory. This novel reading route avoids the fragile tip in the conventional HDD and the difficulty in the direct contact of an MTJ with the SAF nanotrack. Our investigation may pave the way to develop a readable AFM racetrack memory with a high reading speed.

## 2. Model

The numerical investigation is based on the model shown in Figure 1. The dimension of the cell for the simulation is 5 nm × 1 nm × 6 nm. We consider an HM1/FM/NM/FM/HM2 multilayer track system. Here, HM1 and HM2 are two different heavy metal layers, and NM is a nonmagnetic metal layer working as the media for the Ruderman-Kittel-Kasuya-Yosida (RKKY)-typed exchange coupling between the two FM layers with PMA. The interlayer RKKY AFM exchange coupling is characterized by a negative exchange coupling constant (*J_ex_*). 

In an HM/FM system, the magnetic parameters such as the uniaxial-anisotropy constant and the DMI constant can be manipulated experimentally. For example, the uniaxial-anisotropy constant can be too small to maintain the PMA or be as large as 10^6^ J/m^3^. The DMI constant can be between 0 and 3 mJ/m^2^ [25], depending on the composition and thickness of the HM or FM layers and the annealing treatment. All the parameters used in our simulation are in an experimentally achievable range. Without loss of generality, we applied a positive DMI constant and a positive spin-Hall angle that ensures the motion of a right-handed DW against the direction of the current (along the direction of electron flow) [15]. The change of the sign of DMI or the spin Hall angle only reverses the motion direction of the DW but does not influence the conclusion of this paper.

In the HM1/FM1/NM/FM2/HM2 system, to ensure the inter-layer AFM coupling between the DW moments, the DMI constants of the HM1/FM1 and FM2/HM2 need to share the same signs. When the DMI constants of both layers are positive, the chirality requires the DWs to exhibit an “up-down” configuration that is favorable for an inter-layer AFM coupling between the DWs of the two layers (Figure 2). To ensure the motion of the two DWs in the same direction, the spins injecting into FM1 and FM2 should have the same sign in principle. Therefore, the signs of the spin Hall angles of HM1 and HM2 needs to be opposite since the HM1 is below FM1 while the HM2 is above FM2. However, when the interlayer AFM coupling is strong, the rotation of the moment in one layer is pinned by that in the other one. Therefore, it is enough that one HM layer has spin Hall effect [18]. Some compositions of HM/FM satisfy the above requirement, such as Pt/Co and Co/Ir/Pt [26], and W/Co and Co/Ru/W [25].

Because of the difference in the uniaxial-anisotropy constants between the two FM layers and a moderate inter-layer exchange coupling, the velocities of the DWs in the two layers can be very different. As a result, there is a region consisting of the staggered domains between the two neighboring DWs in the two layers, and the magnetic moments in this staggered region are parallel aligned. Two orientations (+z and −z) of the moments in the staggered region can be exploited to label the information stored in the neighboring AFM-coupled domains moving in a fixed direction. For example, as shown in Figure 1, when the reading head nearby the track detects the magnetization of +2M_S_ in a staggered domain region, the reader provides an output voltage of +V. When the domain walls move left, the AFM-coupled domains following this staggered region has the magnetization of −M_S_ for the upper layer and +M_S_ for the lower one. This sort of AFM coupling can be encoded as 0. On the contrary, the AFM-coupled domains with the magnetization of +M_S_ for the upper layer and −M_S_ for the lower one can be encoded as 1. This reading function can be realized by connecting a Schmitt trigger to the sensor. In the track with the domains moving left, when the reader gives an output of +V, a digital signal 1 is triggered, and it holds for the following AFM-coupled domains till the sensor detects the magnetization of −2M_S_. Afterwards, the sensor outputs a voltage of −V that changes the digital signal to 0.

## 3. Results and Discussion

### 3.1. The Results of the Micromagnetic Simulation

The domains and DWs in the track composed by two FM layers with different uniaxial anisotropy energies can be driven by SOT. The representative snapshots of the moving DWs at different times are shown in Figure 2. The magnetic parameters for the simulation are *J_ex_* = −0.15 × 10^−3^ J/m^2^, and the DMI constant D = 0.75 mJ/m^2^ for both layers, and the uniaxial anisotropy constants for the upper and the lower layers are K_U_ = 6 ×10^5^ J/m^3^ and K_L_ = 2.5 ×10^5^ J/m^3^, respectively. In the initial state, a pair of DWs was located in the middle of the bilayer track. (The track region on the right of the paired DWs was not shown.) One can see an approximate AFM coupling between the moments of the two layers in the initial state in the enlarged figure. However, the lengths of the DWs in the two layers are somewhat different due to their distinct uniaxial anisotropy constants [27]. After the current was injected along the *x* direction, the DWs in the two layers moved to the left (the −*x* direction) at different speeds. As a result, when *t* = 0.5 ns, a staggered domain region with a length of about 40 nm (the distance between the centrals of the two DWs) was formed. An obvious parallel alignment of the moments in the two FM layers was shown. When *t* = 1 ns, the length of the staggered domain region increased to 50 nm, and this length nearly unchanged from *t* = 1 ns to *t* = 10 ns. This indicates that the velocity difference of the DWs in the two layers occurred in the early stage of the DW motion. When the DW motion became stable, the gap of the velocity between the two layers disappeared, and the length of the staggered domain region was approximately constant. 

It is also noticed that the structures of the DWs in the two layers are not the same. In the initial state, the DWs in both FM layers exhibited a Néel-typed structure. When the DW was driven by the SOT, the DW in the upper layer that moved slower still exhibited a Néel-typed structure, but in the lower one, the DW that moved faster had a Bloch-typed one.

In Figure 3, the DW motion as a function of D are depicted (Other parameters of *J_ex_*, K_L_, and K_U_ were fixed as −0.3 mJ/m^2^, 2.5 × 10^5^ J/m^3^, and 6 × 10^5^ J/m^3^). The solid and dashed lines in Figure 3a show the temporal central positions of the DWs in the upper and lower track, respectively. When the D varied between 0.1 and 2.0 mJ/m^2^, the solid and dashed lines were overlapped for all the D values but expect D = 0.1 and 0.25 mJ/m^2^. The lengths of the staggered domain region (L_s_) for D = 0.1 and 0.25 mJ/m^2^ were as small as 5 and 10 nm (1 and 2 cells), respectively. The average and the instantaneous velocity at 10 ns are depicted in Figure 3c,d, respectively. For all the D values, the DW velocities for the upper layer are almost identical to that for the lower one, showing that the DWs in the two layers moved at almost the same speeds.

The DW motion for different *J_ex_* is shown in Figure 4. When the *J_ex_* was weaker than −0.25 mJ/m^2^, the DW motion in the upper layer was slower than that in the lower layer. On the other hand, when the *J_ex_* was −0.75 mJ/m^2^, the DW in the upper layer moved together with that in the lower layer even though the anisotropy constants of the two layers are very different. In between, a stable staggered domain region can be generated, and the L_s_ increased with the decreasing *J_ex_*. The stability for the staggered DW region is confirmed from the average velocity and the instantaneous velocity at 10 ns (Figure 4c,d). When the *J_ex_* was between −0.25 mJ/m^2^ and −0.75 mJ/m^2^, there was a small difference in the average velocities for the DWs in the two layers, yet the instantaneous velocities at 10 ns for the two DWs were the same. This means that the velocities of the two DWs were different only in the initial stage of the DW motion. It disappeared later on, and both DWs moved together at the same speed with a stable staggered domain region eventually.

The influence of the difference of uniaxial anisotropy constants between the upper and lower layers on the motion of the two DWs is described in Figure 5. The K_L_ was fixed as 2.5 × 10^5^ J/m^3^, and the K_U_ was between 3 × 10^5^ J/m^3^ and 8 × 10^5^ J/m^3^. When the K_up_ was 6 × 10^5^ J/m^3^ or larger, both the average and instantaneous velocities for the upper and lower DWs were quite distinct (Figure 5c,d), which means that the L_s_ kept increasing in the process of the DW motion (Figure 5b). However, when the K_U_ was between 3 ×10^5^ J/m^3^ and 5 ×10^5^ J/m^3^, the average velocities for the upper and lower DWs were different but both DWs shared the same instantaneous velocities at 10 ns. Therefore, a stable staggered domain region was generated when the K_U_ was 5 × 10^5^ J/m^3^ or smaller. The L_s_ were as large as 50 nm (ten cells) when the K_U_ was 5 × 10^5^ J/m^3^.

For the chiral DW in an ultrathin FM film with a DMI and PMA, the SOT itself cannot drive the DW but can only rotate the DW moment, leading to the initial transition from a Néel DW to a Bloch one. In this process, the projection of moments in the *y* direction increased. Due to the DMI effective field along the *x* direction, there was a DMI torque acting on the DW moment along the *z* direction. This torque rotated the DW moment and pushed the DW to move [2].

After the initial generation of the Bloch DW by the SOT, the following DW motion is factually a transmission of the moment rotation between *z*-axis direction and *y*-axis direction along the length direction of the track. For out-of-plane moments outside the DW, the uniaxial magnetic anisotropy energy offers a barrier for their rotation. In the layer with a smaller uniaxial anisotropy constant, these moments are easier to be rotated, and the DW structure is closer to the Bloch type (Figure 2). As a result, the angle between the DW moment and the DMI field is closer to 90 degrees, which led to a stronger DMI torque. Therefore, the Bloch DW with weak anisotropy energy moved faster than the Néel-typed one with stronger anisotropy energy. 

The important influence of DMI on the DW velocity can be confirmed in Figure 6. Here, the DMI constant for the lower layer (D_L_) was fixed as 0.75 mJ/m^2^, and that of the upper layer (D_U_) varied from 0.05 to 0.75 mJ/m^2^. When the D_U_ was between 0.45 and 0.65 mJ/m^2^, the L_s_ almost kept constant when the DWs moved from 1 to 10 ns. When the two layers shared the same DMI constants, the length of the staggered domains at 1 ns was also very close to that at 10 ns. However, when the D_U_ was smaller than 0.45 mJ/m^2^, the enlarged difference between D_U_ and D_L_ resulted in an unstable L_s_. Initially (*t* = 1 ns), the staggered region was much longer. However, it became shorter later on (*t* = 10 ns). This result indicates that the upper DW moved much faster than the lower one initially, but the lower DW caught up with the upper one later. Even though the anisotropy energy for the lower DW is stronger, its higher DMI can drive the DW to accelerate and catch up with the upper one.

As to two isolated HM/FM nanotracks without any interlayer exchange coupling, the difference of the anisotropy constants between them will lead to an eternal velocity gap and an ever-increased L_s_. In the present work, however, the difference of velocity only occurred in the initial stage of the DW motion in most cases, and it disappeared quickly, resulting in a stable staggered domain region with a fixed L_s_. This is because of the interlayer exchange coupling, which is another important factor influencing the DW velocity in a SAF bilayer [2,18].

The interlayer exchange coupling depends on *J_ex_*, and the angle and the distance of the moments in the two layers. The interlayer exchange coupling in this work is complicated since the DWs in the two layers are staggered. However, the generation of a staggered region with a fixed length may be understood from the aspect of energy competition roughly. Here we focused on the interaction between the moment in one layer and its nearest moment in the other layer (“inter-moment” coupling) and that between the moments in the staggered DWs of the two layers (“inter-DW” coupling). In the “inter-moment” case, the increase in the L_s_ leads to an enhancement of the exchange energy in the staggered region due to the contradiction between the parallel alignment of the moments in this region and the negative *J_ex_*. On the other hand, the inter-DW coupling keeps being reduced during the increase of the length of the staggered region. The competition between the above two sorts of exchange coupling tunes the difference of the velocity between the two DWs and eventually leads to an identical DW velocity and a constant L_s_.

### 3.2. Theoretical Analysis Based on the Cooperative Coordinate Method

The influence of the inter-DW coupling has been quantitatively estimated based on the cooperative coordinate method (CCM). For simplicity, we only considered the “inter-DW” coupling as shown in Figure 7. We have derived a group of Thiele equations depicting the dynamic behavior of the central coordinates (*q*) and the azimuthal angle (*φ*) of the upper and lower layers (The more specific information about the derivation of these equations are depicted in detail in the Appendix A) [2,18,28,29].

(1)$$\frac{\alpha_{L}}{\Delta_{L}}{\dot{q}}_{L}+{\dot{\varphi }}_{L}=\frac{\pi \gamma_{0}H_{SO}J\cos\varphi_{L}}{2}+\frac{2\gamma_{0}\min(\Delta_{L},\Delta_{U})J_{ex}(q_{L}-q_{U})}{\mu_{0}M_{L}{\left[t_{s}^{2}+{\left(q_{L}-q_{U}\right)}^{2}\right]}^{\frac{3}{2}}}$$

(2)$$\frac{{\dot{q}}_{L}}{\Delta_{L}}-\alpha_{L}{\dot{\varphi }}_{L}=\frac{\gamma_{0}\pi D_{L}\sin(\varphi_{L})}{2\Delta \mu_{0}M_{L}}+\frac{\gamma_{0}N_{x}^{L}M_{L}\sin(2\varphi_{L})}{2}+\frac{2\gamma_{0}\min(\Delta_{L},\Delta_{U})J_{ex}\sin(\varphi_{L}-\varphi_{U})}{\mu_{0}\Delta_{L}M_{L}{\left[t_{s}^{2}+{\left(q_{L}-q_{U}\right)}^{2}\right]}^{\frac{1}{2}}}$$

(3)$$\frac{\alpha_{U}}{\Delta_{U}}{\dot{q}}_{U}-{\dot{\varphi }}_{U}=-\frac{\pi \gamma_{0}H_{SO}J\cos\varphi_{U}}{2}-\frac{2\gamma_{0}\min(\Delta_{L},\Delta_{U})J_{ex}(q_{L}-q_{U})}{\mu_{0}M_{L}{\left[t_{s}^{2}+{\left(q_{L}-q_{U}\right)}^{2}\right]}^{\frac{3}{2}}}$$

(4)$$\frac{{\dot{q}}_{U}}{\Delta_{U}}+\alpha_{U}{\dot{\varphi }}_{U}=\frac{\gamma_{0}\pi D_{U}\sin(\varphi_{U})}{2\Delta_{U}\mu_{0}M_{U}}-\frac{\gamma_{0}N_{x}^{U}M_{U}\sin(2\varphi_{U})}{2}+\frac{2\gamma_{0}\min(\Delta_{L},\Delta_{U})J_{ex}\sin(\varphi_{L}-\varphi_{U})}{\mu_{0}\Delta_{U}M_{U}{\left[t_{s}^{2}+{\left(q_{L}-q_{U}\right)}^{2}\right]}^{\frac{1}{2}}}$$

Here, Δ, *α*, *D*, *μ*_0_, *γ*_0_, *M*, *N_x_*, *J_ex_*, *t_s_*, *H_SO_*, and *J* are the width of domain wall, the damping coefficient, the DMI constant, the vacuum permeability, the Gilbert gyromagnetic ratio, the saturation magnetization, the demagnetization factor, the interlayer exchange coupling constant, the distance between the two FM layers, the effective magnetic field of SOT, and the current density. The subscripts *L* and *U* represent the lower and the upper layers.

Figure 7a shows the motion of the paired DWs with the same anisotropy constants (K_L_ = K_U_ = 6 × 10^5^ J/m^3^) and a moderate interlayer exchange coupling (*J_ex_* = −0.3 mJ/m^2^). Figure 7c shows that with different anisotropy constants of the two layers (K_L_ = 2 × 10^5^ J/m^3^ and K_U_ = 6 × 10^5^ J/m^3^) and a strong interlayer exchange coupling (*J_ex_* = −0.75 mJ/m^2^). One can see that in both cases the L_s_ was negligible. However, when the anisotropy constants of the two layers were different (K_L_ = 2 × 10^5^ J/m^3^ and K_U_ = 6 × 10^5^ J/m^3^) and the interlayer exchange coupling was moderate (*J_ex_* = −0.3 mJ/m^2^), a stable staggered domain region with a fixed length of about 40 nm was generated (Figure 7b). This result is consistent with the result of the micromagnetic simulation.

### 3.3. The Application of the Staggered Domain Region in the SAF Racetrack Memory

In a real racetrack memory, a short current pulse induces the motion of the DWs. Therefore, the relaxation of the DW motion after turning on or off the current needs to be considered. Figure 8 indicates the variation of the L_s_ with an injected current pulse. From the snapshot shown in Figure 8a, one can see that after turning off the current, the staggered region experienced an oscillation attenuation for around 0.5 ns and disappeared finally. When a current was switched on, the staggered region was generated also after a 0.5-ns relaxation. 

The evolution of the L_s_ with the injection of a current pulse under different conditions is shown in Figure 8c–e. One can see that the relaxation after the current pulse was switched on or off contributes to a short rising or falling edge in the temporal L_s_. For a large damping coefficient (*α* = 0.1), a large difference of K_L_ and K_U_ (K_L_ = 2.5 × 10^5^ J/m^3^, K_U_ = 6 × 10^5^ J/m^3^) can still generate a stable staggered domain region with an L_s_ as large as 90 nm (Figure 8d). In this case, the duration of the rising and falling edge can be shorter than 0.5 ns, which is much smaller than the period of the pulse in current electronic devices. A small increase in the interlayer exchange coupling reduced the L_s_ significantly, but it did not influence the time for the rising and falling edge (Figure 8c). One the other hand, the time for the rising edge is very close to that of the falling edge. A detailed depiction of the duration of rising and falling edges as a function of magnetic parameters is described in the Appendix A.

In addition to the dynamic behavior of the staggered domain region, the minimum interval between two neighboring DWs also needs to be considered since it determines the storage density. Even though the net magnetization of the paired domains is zero, the interval between the chiral DWs cannot be infinitely small because of the magnetostatic interaction between the neighboring DWs with the same chirality.

As shown in Figure 9a, an array of domains with an interval of 25 nm was set initially. After the relaxation, the DWs were generated and the length of the domain was modified. The middle red domain in the upper layer became much shorter than its neighbors due to the magnetostatic interaction from the red domains on its two sides. Two red domains with very distinct lengths are unstable in energy, and they are finally merged into a long red domain by “swallowing” the blue domain in between after they were induced to move. The same phenomenon was observed for the domains in the lower layer with a much smaller anisotropy constant. 

When the initial DW interval increased to 50 nm (II), the lengths of the middle and right domains in the upper layer are very close after the initial relaxation. After the domains were induced to move, the lengths of these domains did not change. Therefore, the domains are safe in the process of the DW motion when the initial interval between the DWs is 50 nm or larger.

A bilayer track with a series of DWs with different initial intervals (50 nm or larger (III)) was used as a typical example. When the DWs started moving, the magnetization orientation in the staggered domain region can be measured by detecting the stray field using a sensor near the track. Figure 9b depicts the output of the sensor. Here the +V (−V) corresponds to the magnetization of +2M_S_ (−2M_S_) in the staggered domain region. For the DWs moving left, the +V (−V) output indicates the magnetization of the following AFM-coupled domains is −M_S_ (+M_S_) for the upper layer and +M_S_ (−M_S_) for the lower one, and the binary information 0 (1) was encoded accordingly. Thanks to the Schmitt Trigger connected to the sensor, the information stored in the AFM-coupled domains can be finally determined (Figure 9c). Here, the size for a memory unit is the shortest length of the domain that is determined by the allowed smallest interval between the two DWs. It is noticed that when the initial domain length was set as 50 nm or larger, the domain length was very close to its initial set value after the DWs started moving.

## 4. Summary

To resolve the difficulty in reading the information stored in the AFM-coupled domains in a SAF racetrack system, a SAF multilayer which is composed of two FM layers with different anisotropy constants was proposed. Due to the difference in the anisotropy constants between the two FM layers, a staggered domains region with the parallel alignment of magnetization was generated. For a moving DW in a fixed direction, the orientation of the moments in this staggered region can be exploited to label the structure of the following AFM-coupled domains and offers the possibility to read the digital information stored in the AFM-coupled domains. In a real racetrack memory, this reading function can be realized by connecting a Schmitt Trigger to a reading sensor. Through manipulating the magnetic parameters including the interlayer exchange coupling constant, the DMI constants and the anisotropy constants of the two FM layers, the stability, the relaxation process, and the length of the staggered domain regions can be well tuned.

## 5. Method

In the present work, the motion of the DWs was driven by a current. In general, four terms, the adiabatic and the non-adiabatic terms of STT and the field-like and the damping-like terms of SOT may be contributed to the Landau-Lifshitz-Gilbert (LLG) equation that depicts the motion of the DWs. The LLG equation is expressed as:(5)$$\frac{\partial \vec{m}}{\partial t}=-\gamma_{0}\vec{m}\times {\vec{H}}_{\mathrm{eff}}+\alpha (\vec{m}\times \frac{\partial \vec{m}}{\partial t})+\gamma_{}H_{\mathrm{SO}}(\vec{m}\times (\vec{\sigma }\times \vec{m}))$$

Here *γ*_0_, *α*, $\vec{m}$, *t*, *γ*, and $\vec{\sigma }$ are the Landau-Lifshitz gyromagnetic ratio, the Gilbert damping coefficient, the unit vectors for the direction of magnetization, time, gyromagnetic ratio, and the unit vector for spin orientation, respectively. ${\vec{H}}_{\mathrm{eff}}$ is the effective magnetic field related to the free energy density (*E*) by:(6)$${\vec{H}}_{\mathrm{eff}}=-\frac{1}{\mu_{0}M_{S}}(\frac{\delta E}{\delta \vec{m}})$$

Here *μ*_0_ and *M*_S_ are the vacuum permeability and the saturation magnetization, respectively. ${\vec{H}}_{\mathrm{eff}}$ includes the effective fields contributed from exchange coupling in every FM layer, the interlayer RKKY exchange coupling, the magnetic anisotropy energy, the demagnetization energy, and the DMI. *H_SO_* is the effective field of the SOT, and it is expressed as [15]:(7)$$H_{\mathrm{SO}}=\mu_{B}\theta_{\mathrm{SH}}J/\gamma_{0}eM_{S}L_{z}$$

Here *μ*_B_, *θ*_SH_, *J*, and e are the Bohr magneton, the spin Hall angle of the HM layer, the current density, and the charge of an electron, respectively. The parameter *γ*_0_ is related to *γ* by $\gamma_{0}=\mu_{0}\left|\gamma \right|$.

The Equation (1) was numerically solved by the software named “object-oriented micromagnetic framework” (OOMMF) with a code including DMI, SOT, and RKKY-style inter-layer exchange coupling [30]. We do not consider the bulk STT contribution since it is reported that this contribution is usually marginal [2,9,27,30,31]. We have compared the contributions of the STT and SOT in the Appendix A. One can see that the DW velocity from the STT is indeed much lower than that from the SOT under the same current density.

## Figures and Tables

**Figure 1 nanomaterials-09-01538-f001:**
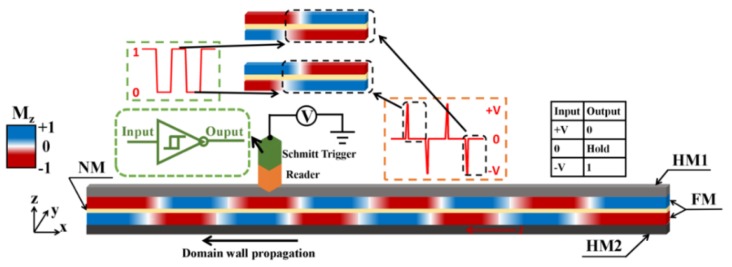
The schematic of the model and principle for the readable SAF racetrack memory.

**Figure 2 nanomaterials-09-01538-f002:**
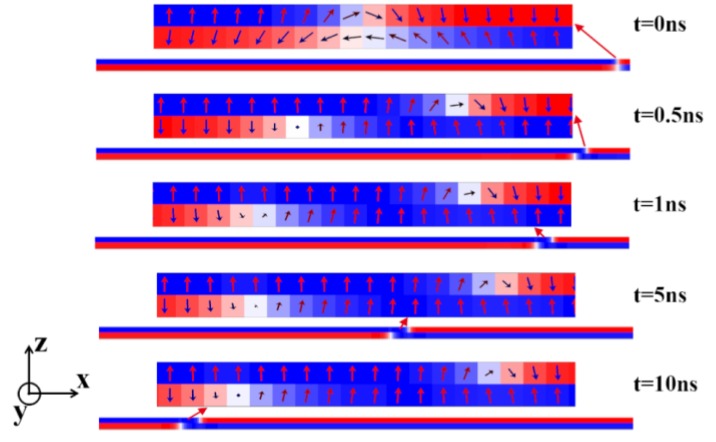
The representative snapshots of the position of the DW and the enlarged figures for the DW structure in the SAF racetrack memory at different times. The magnetic parameters of *J_ex_*, D, K_U_, and K_L_ were fixed as −0.15 mJ/m^2^, 0.75 mJ/m^2^, 5 × 10^5^ J/m^3^, and 2.5 × 10^5^ J/m^3^.

**Figure 3 nanomaterials-09-01538-f003:**
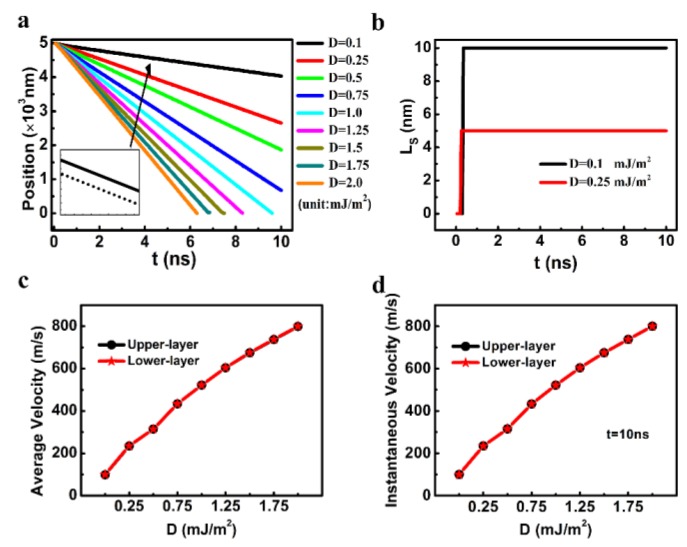
(**a**) The temporal central positions of the magnetic domain walls (DWs) in the two layers (The solid and dashed lines represent the results in the upper layer and in the lower one, respectively.), (**b**) the temporal length of the staggered domains of the synthetic antiferromagnets (SAF) racetrack, (**c**) the average velocity, and (**d**) the instantaneous velocity of the DW as a function of the Dzyaloshinskii–Moriya interaction (DMI) constant. Other parameters of *J_ex_*, K_L_, K_U_ were fixed as −0.3 mJ/m^2^, 2.5 ×10^5^ J/m^3^, and 6 ×10^5^ J/m^3^, respectively.

**Figure 4 nanomaterials-09-01538-f004:**
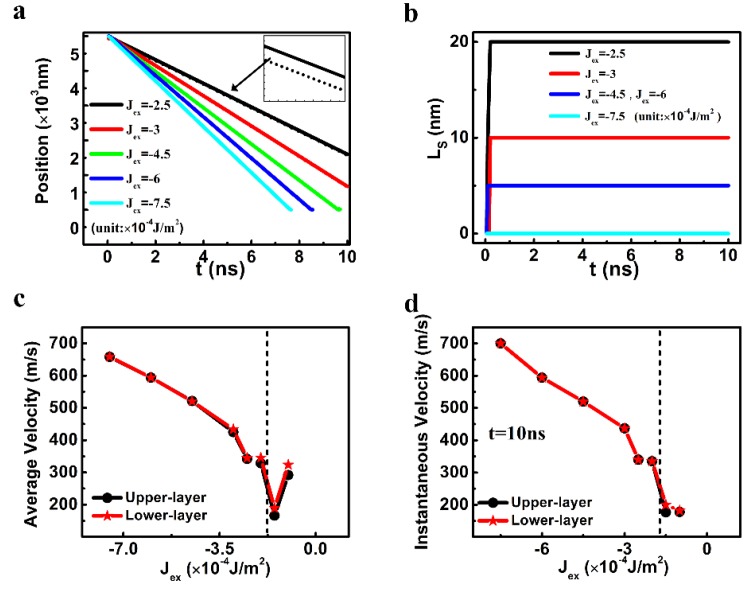
(**a**) The temporal central positions of the DWs in the two layers (The solid and dashed lines represent the results in the upper layer and in the lower one, respectively.), (**b**) the temporal length of the staggered domains of the SAF racetrack, (**c**) the average velocity, and (**d**) the instantaneous velocity of the DW as a function of the interlayer exchange coupling constant. Other parameters of D, K_L_, K_U_ were fixed as 0.75 mJ/m^2^, 2.5 × 10^5^ J/m^3^, and 6 × 10^5^ J/m^3^, respectively.

**Figure 5 nanomaterials-09-01538-f005:**
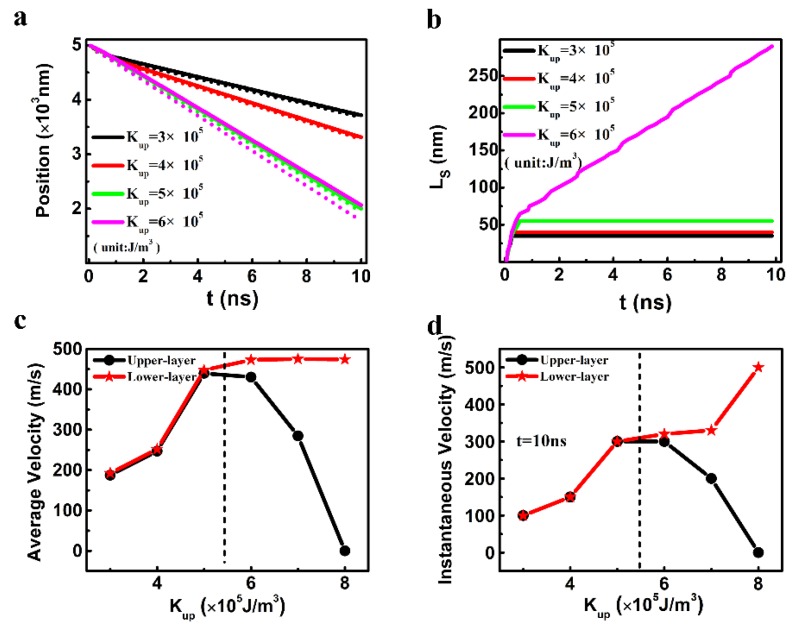
(**a**) The temporal central positions of the DWs in the two layers (The solid and dashed lines represent the results in the upper layer and in the lower one, respectively.), (**b**) the temporal length of the staggered domains of the SAF racetrack, (**c**) the average velocity, and (**d**) the instantaneous velocity of the DW as a function of the anisotropy constant of the upper layer. Other parameters of *J_ex_*, D, and K_L_ were fixed as −0.15 mJ/m^2^, 0.75 mJ/m^2^, and 2.5 × 10^5^ J/m^3^, respectively.

**Figure 6 nanomaterials-09-01538-f006:**
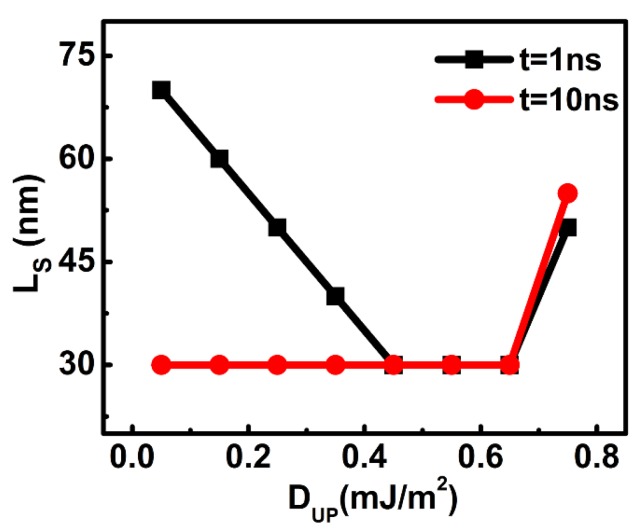
The length of the staggered domain region as a function of the DMI constant of the upper layer at different times. (The K_U_ and K_L_ were fixed as 2.5 × 10^5^ J/m^3^ and 5 × 10^5^ J/m^3^, and *J_ex_* = −0.15 mJ/m^2^, and D_L_ = 0.75 mJ/m^2^).

**Figure 7 nanomaterials-09-01538-f007:**
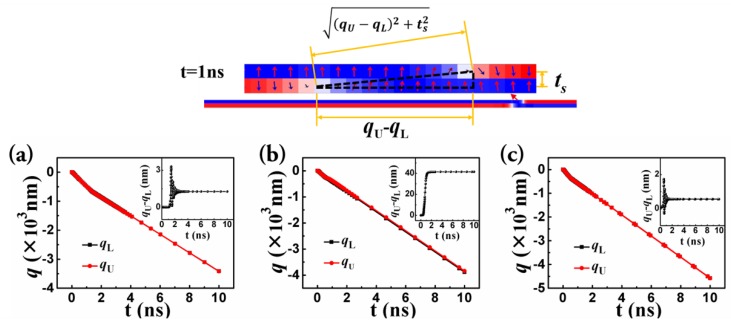
The evolution of the central positions of the lower layer (*q_L_*) and that of the upper layer (*q_U_*) for (**a**) K_L_ = K_U_ = 6 × 10^5^ J/m^3^, and *J_ex_* = −0.3 mJ/m^2^, (**b**) K_L_ = 2 × 10^5^ J/m^3^ and K_U_ = 6 × 10^5^ J/m^3^, and *J_ex_* = −0.3 mJ/m^2^, and (**c**) K_L_ = 2 × 10^5^ J/m^3^ and K_U_ = 6 × 10^5^ J/m^3^, and *J_ex_* = −0.75 mJ/m^2^. The upper panel indicates the inter-DW exchange coupling that was considered in the CCM.

**Figure 8 nanomaterials-09-01538-f008:**
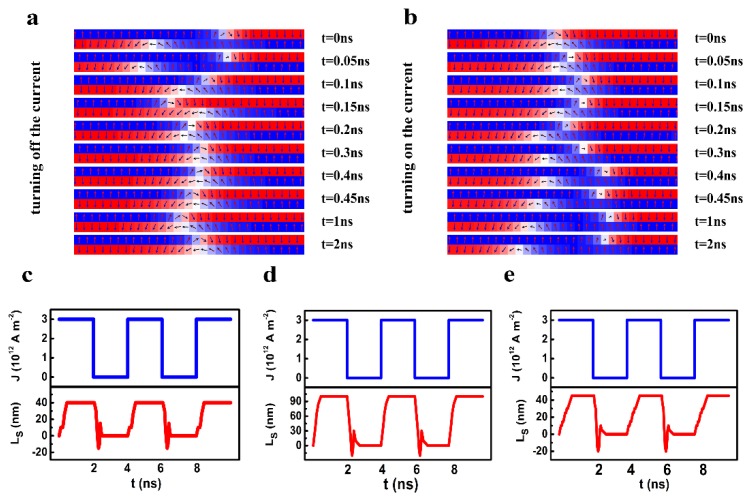
The snapshots of the decaying oscillation of the staggered domain region (**a**) after turning off the current and (**b**) after turning on the current (*α* = 0.1, *J_ex_* = −0.2 × 10^−3^ J/m^2^, D = 0.25 mJ/m^2^, K_up_ = 6 × 10^5^ J/m^3^, K_low_ = 2.5 × 10^5^ J/m^3^), (**c–e**). the evolution of the L_s_ with the injection of the current pulse under different magnetic parameters ((**c**): *α* = 0.1, *J_ex_* = −0.2×10^−3^ J/m^2^, D = 0.25 mJ/m^2^, K_up_ = 6 × 10^5^ J/m^3^, K_low_ = 2.5 × 10^5^ J/m^3^, (**d**): *α* = 0.1, *J_ex_* = −0.15 × 10^−3^ J/m^2^, D = 0.75 mJ/m^2^, K_up_ = 6 × 10^5^ J/m^3^, K_low_ = 2.5 × 10^5^ J/m^3^, (**e**): *α* = 0.1, *J_ex_* = −0.2 × 10^−3^ J/m^2^, D = 0.75 mJ/m^2^, K_up_ = 4 × 10^5^ J/m^3^, K_low_ =2.5 × 10^5^ J/m^3^).

**Figure 9 nanomaterials-09-01538-f009:**
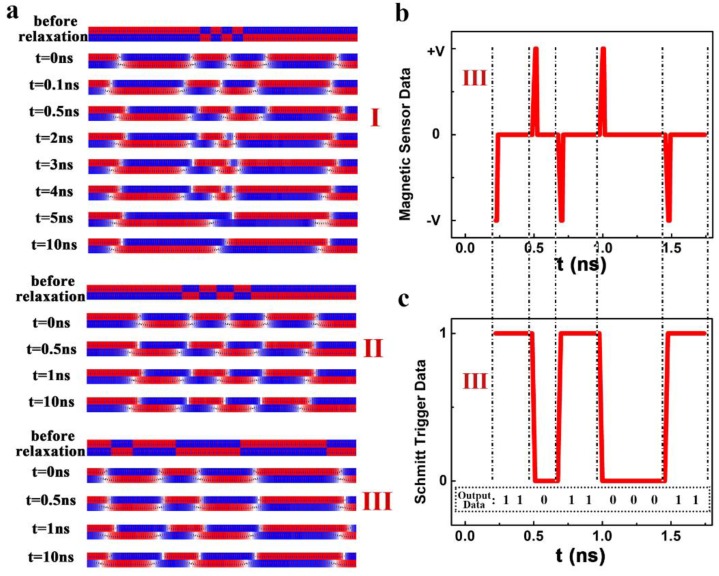
(**a**) The DW structure of the SAF racetrack at a different time with different initial intervals of the domain, (**b**) the output of the magnetic sensor and (**c**) the related digital signals for the case III in Figure 6a. The magnetic parameters including *J_ex_*, *D*, K_U_, and K_L_ were fixed as −0.25 mJ/m^2^, 0.75 mJ/m^2^, 6 × 10^5^ J/m^3^, and 2.5 × 10^5^ J/m^3^, respectively.

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
