# Peer review of "Readable High-Speed Racetrack Memory Based on an Antiferromagnetically Coupled Soft/Hard Magnetic Bilayer"

_nanomaterials, 2019, doi:10.3390/nano9111538_

Round 1

Reviewer 1 Report

Report – 10th of October 2019

Title of the Article: “ Readable high-speed racetrack memory based on an antiferromagnetically coupled soft/hard magnetic bilayer” (nanomaterials-616139)

Authors: Ziyang Yu, Chenhuinan Wei, Fan Yi, Rui Xiong

The paper reports on the current-induced domain wall motion in a racetrack memory. Authors report on the challenging task of reading the information stored in domains linked to the zero net magnetization in the case of antiferromagnetically coupled domains. They propose to solve this issue by using a two ferromagnetic layers based memory. Thanks to the different uniaxial anisotropy constants of the two ferromagnetic layers, they are able to demonstrate that a parallel alignment is reachable (in a staggered region) and thus easily readable. This allows labeling the nearby antiferromagnetic domains by a comparator circuit (Schmitt trigger) connected to a reading sensor. They also study the stability and the length of the parallel-aligned domains as function of the magnetic characteristics of the ferromagnetic layers.

This is a very interesting work for future applications of ferromagnetic nanostructures in new high-speed memory. Nevertheless some minor points have still to be considered before publication:

The English of the article should be examined for the last version of the paper. The general context is, the most of the time, clear but few sentences are inaccurate. Sometimes verbs are forgotten, sometimes syntax sounds weird… It is regrettable that only little information is given for the micromagnetic simulations done. In the method section is only reported that equations were solved by OOMMF, it is not clear if the same framework was used for the simulation of the bi-layered system and which kind of discretization/size has been used for the nanostructures. The supplementary information is unreadable and thus impossible to review. All the equations present problems of symbols.

In conclusion, I found that the article worth for publication after minor revisions aiming a clear presentation of the interesting results.

Author Response

Response to Reviewer 1 Comments

Point 1: The English of the article should be examined for the last version of the paper. The general context is, the most of the time, clear but few sentences are inaccurate. Sometimes verbs are forgotten, sometimes syntax sounds weird… It is regrettable that only little information is given for the micromagnetic simulations done. In the method section is only reported that equations were solved by OOMMF, it is not clear if the same framework was used for the simulation of the bi-layered system and which kind of discretization/size has been used for the nanostructures. The supplementary information is unreadable and thus impossible to review. All the equations present problems of symbols.

Respond 1: We do appreciate the reviewer’s critical comments on our work.

I’m sorry that you can’t see supplementary information because of the file’s format issues. We have uploaded the supplementary material in submit system and attachment. Please see the attachment.

Reviewer 2 Report

This manuscript reports a theoretical and numerical study on coupled domain wall dynamics in a synthetic antiferromagnet (SAF). It is shown that dissimilar magnetic properties of two ferromagnetic layers result in lateral displacement of two domain walls. The results in this manuscript seem technically sound. I support publication of this work in Nanomaterials once the authors revise the manuscript for the following comments:

In the introduction, it was mentioned that employing tunnel magnetoresistance (TMR) and MTJ as a reading scheme is not compatible with the fabrication of a capping layer. I disagree. Using MTJ and TMR is essential for practical application of racetrack memory and, for instance, a MgO layer can be directly deposited on top of SAF without breaking vacuum. In this respect, a lateral displacement of two domain walls during dynamics is not necessary for reading. Please modify the introduction and also the title. The main result of this work is similar to Nat. Phys. 15, 543 (2019), which reports chiral exchange drag. Please add a discussion about similarity and difference with this NPhys paper.

Author Response

Response to Reviewer 2 Comments

Point 1: In the introduction, it was mentioned that employing tunnel magnetoresistance (TMR) and MTJ as a reading scheme is not compatible with the fabrication of a capping layer. I disagree. Using MTJ and TMR is essential for practical application of racetrack memory and, for instance, a MgO layer can be directly deposited on top of SAF without breaking vacuum. In this respect, a lateral displacement of two domain walls during dynamics is not necessary for reading. Please modify the introduction and also the title.

Respond 1:We do appreciate the reviewer’s critical comments on our work.

Employing tunnel magnetoresistance (TMR) and MTJ as a reading scheme is compatible with the fabrication of a capping layer. And TMR and MTJ is essential to read the signal in racetrack memory. I agree with all these. The difficulty of the reading in a SAF racetrack memory is a truth that is generally accepted. On the other hand, in a real SAF racetrack, the uppermost layer is not a FM layer but a capping layer to avoid the oxidation. Therefore, a direct contact of the reading sensor with the upper FM layer is quite difficult in experiments. Additionally, we do not think this stray field in the SAF racetrack can be easily detected since the two FM layers are very close (their spacing is 1 nm or smaller). In this case, the stray field is quite negligible no matter the reading sensor is in contact with the wire or not.

Point 2: The main result of this work is similar to Nat. Phys. 15, 543 (2019), which reports chiral exchange drag. Please add a discussion about similarity and difference with this NPhys paper.

Respond 2:This Npnys paper investigated that domain walls (DWs) can be driven by current via chiral spin-orbit torques. They found that the DW in one sublayer of the SAF can drag the DW in the other sublayer via an exchange coupling between. Then they also investigate dynamics and background physical mechanism. Our main result maybe looks similar with their work. These two work both investigate coupled DWs which are under interlayer RKKY exchange coupling are driven by current via chiral spin-orbit torques. But there are many essential differences between these two work. Firstly, we mainly focus on stability and the length of the staggered region can be well tuned by changing the magnetic parameters, such as the interlayer exchange coupling constants, the Dzyaloshinskii–Moriya interaction (DMI) constants, and the uniaxial-anisotropy constants of the two FM layers, in a range that is experimentally achievable. But Parkin group mainly change the ratio mU/mL(where mL and mU are the magnetic moments in two layers), current density and DMI (they only study CED dynamics as a function low layer DMI). Secondly, the theoretical analysis we have studied based on the cooperative coordinate method (CCM) which is different with the mechanism of chiral exchange drag (CED). And the result from our theoretical analysis CCM is consistent with the result of the micromagnetic simulation. Lastly, we have designed the model for the readable SAF racetrack memory and the reading function can be realized by connecting a Schmitt Trigger to a reading sensor.

Reviewer 3 Report

The manuscript submitted by Z. Yua, C. Weia, F. Yib, and R. Xionga deals with an important topic i.e. study a new type of non-volatile magnetic memory based on the current-induced domain walls motion in a two antiferromagnetically coupled thin ferromagnetic layers with a perpendicular magnetic anisotropy. This is attractive for high-speed magnetic data storage and spintronic devices with a high-density magnetic recording. It is a matter of considerable scientific and practical interest.

The authors theoretically studied the domain structure and dynamic properties of multilayer thin films, which consist with pair of the ferromagnetic/hard metal layers divided by non-magnetic spacer using computer simulation. They proposed a model for the readable synthetic antiferromagnet racetrack memory and demonstrated its principle functionality due to various values of magnetic anisotropy in ferromagnetic layers and the Dzyaloshinskii–Moriya interaction and spin-Orbit torque near the ferromagnetic/hard metal interface.

I find the study and analyzing of the staggered domains region with a parallel alignment of magnetization in antiferromagnetically coupled ferromagnetic interesting. I think that the paper is suitable for publication in Nanomaterials with some revision.

In the following I will list a few issues and comments that should be addressed to the revised paper.

- Because Thiele equations is not wide known for the majority of the scientific community, it is necessary to insert in the text the reference with the description of these equations.

- The reference [27] is not correct. Nothing is about using the OOMMF software in this article.

- It is not clear way in figs. 3a, 4a, and 5a “the temporal central positions of the DWs in the two layers” starts from 5 nm, but not 0 nm,

- The term “AFM domains” used in abstract is incorrect because antiferromagnet materials are not used in this paper.

There are a few misprints and incorrect phrases in the manuscript:

- no any definition of an abbreviation “DMI” in abstract

- line 70: the phrase “pave a way” is used instead standard one “pave the way”.

- lines 144-145: The phrase “The details … are depicted in detail…” is not correct.

- lines 151-153: What means “but D = 0.1 and 0.25 mJ/m2”in the end of phrase “When the D varied between 0.1 mJ/m2 and 2.0 mJ/m2, the solid and dashed lines were overlapped for all the D values but D = 0.1 and 0.25 mJ/m2”?

- lines 154 and 189: it is not clear what means “cells”.

- line 253: The term γ0 is not defined later in the suggestion.

- The references 25 и 27 are indicated incorrectly.

Author Response

Response to Reviewer 3 Comments

We do appreciate the reviewer’s critical comments on our work.

Point 1: Because Thiele equations is not wide known for the majority of the scientific community, it is necessary to insert in the text the reference with the description of these equations.

Respond 1: I’m sorry that you can’t see supplementary information because of the file’s format issues. We have uploaded the pdf of supplementary material in submit system and attachment. All details about the derivation of these equations are depicted in detail in the Supplementary Materials (S1).

Point 2: The reference [27] is not correct. Nothing is about using the OOMMF software in this article.

Respond 2:Already modified the reference [27].

 “[27] .Donahue MJ, Porter DJ (2002) OOMMF user’s guide, version 1.2a3.“

Point 3: It is not clear way in figs. 3a, 4a, and 5a “the temporal central positions of the DWs in the two layers” starts from 5 nm, but not 0 nm.

Respond 3: This is our negligence that we did not give the particular structure parameters. The structure parameters have been added in our manuscript. The length of nanowire structure is 10×103nm, the initial position of DW is in the middle of nanowire. In addition, the DWs move from 5×103nm to 0nm when the current is along the -x direction.

Point 4: The term “AFM domains” used in abstract is incorrect because antiferromagnet materials are not used in this paper.

Respond 4: Maybe we didn't explain it clearly enough. The term “AFM domains” in abstract can be interpreted as “antiferromagnetically coupled domains”. The “AFM domains” means the two DWs are coupled by interlayer antiferomagnetic (AFM) coupling interaction. In particular, the structure we used in this work can be called “synthetic antiferromagnets (SAFs) structure.

Point 5: no any definition of an abbreviation “DMI” in abstract

Respond 5: The abbreviation of “DMI” have been added in abstract. Line 22: “Dzyaloshinskii–Moriya interaction (DMI)”.

Point 6:  line 70: the phrase “pave a way” is used instead standard one “pave the way”.

Respond 6: Already modified in the manuscript. Line 72: “pave the way”

Point 7: lines 144-145: The phrase “The details … are depicted in detail…” is not correct.

Respond 7: Already modified in the manuscript. Line 247~248: “(The more specific information about the derivation of these equations are depicted in detail in the Supplementary Materials (S1)).”

Point 8: lines 151-153: What means “but D = 0.1 and 0.25 mJ/m2”in the end of phrase “When the D varied between 0.1 mJ/m2 and 2.0 mJ/m2, the solid and dashed lines were overlapped for all the D values but D = 0.1 and 0.25 mJ/m2”?

Respond 8: The solid and dashed lines in Fig. 3a show the temporal central positions of the DWs in the upper and lower track, respectively. The lengths of the staggered domain region (Ls) for D = 0.1 and 0.25 mJ/m2 were as small as 5 and 10 nm (1 and 2 cells), respectively. And the length of the staggered domain region (Ls) is zero when D is bigger than 0.25 mJ/m2. So the solid and dashed lines were overlapped for all the D values but D = 0.1 and 0.25 mJ/m2.

Point 9: lines 154 and 189: it is not clear what means “cells”.

Respond 9: Sorry, we didn’t add the special parameters of structure in our manuscript. Now we have already added the necessary parameters in our manuscript. Line 77~78: “The dimension of the cell for the simulation is 5 nm×1 nm×6 nm. ”

Point 10: line 253: The term γ0 is not defined later in the suggestion.

Respond 10: We have already added the definition of γ0 which represents the Gilbert gyromagnetic ratio. Line 258: “the Gilbert gyromagnetic ratio”.

Point 11: The references 25 и 27 are indicated incorrectly.

Respond 11: Already modified the reference [27].

 “[27] .Donahue MJ, Porter DJ (2002) OOMMF user’s guide, version 1.2a3.”

Round 2

Reviewer 2 Report

The authors agreed with the point 1 in my review report, but did not make any revision in response to this point.

The authors agreed that this manuscript deals a similar physics with NPhy paper, which I mentioned in the point 2, but did not even cite the paper.

On the overall, there is no improvement of the manuscript with respect to my comments. Therefore, I cannot support publication.

Author Response

Response to Reviewer 2 Comments

Point 1: In the introduction, it was mentioned that employing tunnel magnetoresistance (TMR) and MTJ as a reading scheme is not compatible with the fabrication of a capping layer. I disagree. Using MTJ and TMR is essential for practical application of racetrack memory and, for instance, a MgO layer can be directly deposited on top of SAF without breaking vacuum. In this respect, a lateral displacement of two domain walls during dynamics is not necessary for reading. Please modify the introduction and also the title. 

Respond 1:We do appreciate the reviewer’s critical comments on our work.

Employing tunnel magnetoresistance (TMR) and MTJ as a reading scheme is compatible with the fabrication of a capping layer. And TMR and MTJ is essential to read the signal in racetrack memory. I agree with all these. The difficulty of the reading in a SAF racetrack memory is a truth that is generally accepted. On the other hand, in a real SAF racetrack, the uppermost layer is not a FM layer but a capping layer to avoid the oxidation. Therefore, a direct contact of the reading sensor with the upper FM layer is quite difficult in experiments. Additionally, we do not think this stray field in the SAF racetrack can be easily detected since the two FM layers are very close (their spacing is 1 nm or smaller). In this case, the stray field is quite negligible no matter the reading sensor is in contact with the wire or not.

Point 1 (round 2):The authors agreed with the point 1 in my review report, but did not make any revision in response to this point.

Respond 1 (round 2): We do appreciate the reviewer’s critical comments on our work. However, we still hold that our proposal offers a valuable route to resolve the important reading problem in the racetrack memory with a synthetic antiferromagnetic (SAF) structure and what we described in introduction.

In the previous respond, we may not state that very precisely. What we said in our manuscript is “Exploiting the upper FM layer as the free layer of the MTJ is not quite experimentally compatible with the fabrication of a capping layer above the upper FM layer.” in line 61~62. Maybe this sentence is not precise enough. We have revised it to “Exploiting the upper FM layer as the free layer of the MTJ is not quite experimentally compatible with the fabrication of a capping layer above the upper FM layer in SAF structure”. So our opinion is: employing tunnel magnetoresistance (TMR) and MTJ as a reading scheme is not compatible with the fabrication of a capping layer when we use these methods in SAF structure. When we use upper FM layer as the free layer of the MTJ, and the digital information corresponds to the orientation of the moments in the upper FM layer. As for “Employing tunnel magnetoresistance (TMR) and MTJ as a reading scheme is compatible with the fabrication of a capping layer.” We quite agree with this viewpoint. But in the racetrack memory with a synthetic antiferromagnetic (SAF) structure, the uppermost layer is not a FM layer but a capping layer to avoid the oxidation. Therefore, a direct contact of the reading sensor with the upper FM layer is quite difficult in experiments. 

For more details, please refer our revision in the resubmitted manuscript marked as red.(line 58~65)

Point 2: The main result of this work is similar to Nat. Phys. 15, 543 (2019), which reports chiral exchange drag. Please add a discussion about similarity and difference with this NPhys paper.

Respond 2:This Npnys paper investigated that domain walls (DWs) can be driven by current via chiral spin-orbit torques. They found that the DW in one sublayer of the SAF can drag the DW in the other sublayer via an exchange coupling between. Then they also investigate dynamics and background physical mechanism. Our main result maybe looks similar with their work. These two work both investigate coupled DWs which are under interlayer RKKY exchange coupling are driven by current via chiral spin-orbit torques. But there are many essential differences between these two work. Firstly, we mainly focus on stability and the length of the staggered region can be well tuned by changing the magnetic parameters, such as the interlayer exchange coupling constants, the Dzyaloshinskii–Moriya interaction (DMI) constants, and the uniaxial-anisotropy constants of the two FM layers, in a range that is experimentally achievable. But Parkin group mainly change the ratio mU/mL(where mL and mU are the magnetic moments in two layers), current density and DMI (they only study CED dynamics as a function low layer DMI). Secondly, the theoretical analysis we have studied based on the cooperative coordinate method (CCM) which is different with the mechanism of chiral exchange drag (CED). And the result from our theoretical analysis CCM is consistent with the result of the micromagnetic simulation. Lastly, we have designed the model for the readable SAF racetrack memory and the reading function can be realized by connecting a Schmitt Trigger to a reading sensor.

Point 2 (round 2):The authors agreed that this manuscript deals a similar physics with NPhy paper, which I mentioned in the point 2, but did not even cite the paper.

 On the overall, there is no improvement of the manuscript with respect to my comments. Therefore, I cannot support publication.

Respond 2 (round 2): We are very sorry for this negligence. The reason for not quoting this paper before is that we had finished this work before 2019. Anyway, we must cite this paper now. Finally, we do appreciate the reviewer’s suggestion with such a high-quality paper. Now we have already cited this paper in line 56~58.

“[24]Yang S H, Garg C, Parkin S S P. Chiral exchange drag and chirality oscillations in synthetic antiferromagnets[J]. Nature Physics, 2019, 15(6): 543.”

Reviewer 3 Report

Author’s some responses are not clear for me. The paper may be published in Nanomaterials after correct answer the following points.

Point 1: Because Thiele equations is not wide known for the majority of the scientific community, it is necessary to insert in the text the reference with the description of these equations.

Respond 1: I’m sorry that you can’t see supplementary information because of the file’s format issues. We have uploaded the pdf of supplementary material in submit system and attachment. All details about the derivation of these equations are depicted in detail in the Supplementary Materials (S1).

- Authors did not answer this point. Matter of course, I have read the Supplementary Materials, were authors showed the procedure of obtaining the equations Eqs (1)-(4) in the main text using the Lagrange-Rayleigh formalism from S1. However, “Thiele equations” in this reference are not mentioned.

Point 4: The term “AFM domains” used in abstract is incorrect because antiferromagnet materials are not used in this paper.

Respond 4: Maybe we didn't explain it clearly enough. The term “AFM domains” in abstract can be interpreted as “antiferromagnetically coupled domains”. The “AFM domains” means the two DWs are coupled by interlayer antiferomagnetic (AFM) coupling interaction. In particular, the structure we used in this work can be called “synthetic antiferromagnets (SAFs) structure.

- In the scientific literature the " AFM domains " is a polynomial expression, which is associated with domain in antiferromagnet. Therefore, authors should use others words to indicate domains in the antiferromagnetically coupled ferromagnet layers.

Point 8: lines 151-153: What means “but D = 0.1 and 0.25 mJ/m2”in the end of phrase “When the D varied between 0.1 mJ/m2 and 2.0 mJ/m2, the solid and dashed lines were overlapped for all the D values but D = 0.1 and 0.25 mJ/m2”?

Respond 8: The solid and dashed lines in Fig. 3a show the temporal central positions of the DWs in the upper and lower track, respectively. The lengths of the staggered domain region (Ls) for D = 0.1 and 0.25 mJ/m2 were as small as 5 and 10 nm (1 and 2 cells), respectively. And the length of the staggered domain region (Ls) is zero when D is bigger than 0.25 mJ/m2. So the solid and dashed lines were overlapped for all the D values but D = 0.1 and 0.25 mJ/m2.

- may be authors would like say “but except D = 0.1 and 0.25 mJ/m2”.

Point 11: The references 25 и 27 are indicated incorrectly.

Respond 11: Already modified the reference [27].

“[27] .Donahue MJ, Porter DJ (2002) OOMMF user s guide, version 1.2a3.”

- The page in the reference {25] is not indicated.

Author Response

Response to Reviewer 3 Comments

Author’s some responses are not clear for me. The paper may be published in Nanomaterials after correct answer the following points.

Point 1: Because Thiele equations is not wide known for the majority of the scientific community, it is necessary to insert in the text the reference with the description of these equations.

Respond 1: I’m sorry that you can’t see supplementary information because of the file’s format issues. We have uploaded the pdf of supplementary material in submit system and attachment. All details about the derivation of these equations are depicted in detail in the Supplementary Materials (S1).

Point 1 (round 2):  Authors did not answer this point. Matter of course, I have read the Supplementary Materials, were authors showed the procedure of obtaining the equations Eqs (1)-(4) in the main text using the Lagrange-Rayleigh formalism from S1. However, “Thiele equations” in this reference are not mentioned.

Respond 1 (round 2): I’m sorry that we didn’t respond to this point clearly. Now we have added four references (Journal of Physics Condensed Matter 2017, 29 (30), 303001.; Nature Nanotechnology 2015, 10 (3), 221-226.; Journal of Applied Physics, 2018, 123(1):013901.; Physical Review Letters, 1973, 30(6):230-233.) which include more detail information about Thiele equation. I hope these references can support enough information for Thiele equations used in manuscript.

Point 4: The term “AFM domains” used in abstract is incorrect because antiferromagnet materials are not used in this paper.

Respond 4: Maybe we didn't explain it clearly enough. The term “AFM domains” in abstract can be interpreted as “antiferromagnetically coupled domains”. The “AFM domains” means the two DWs are coupled by interlayer antiferomagnetic (AFM) coupling interaction. In particular, the structure we used in this work can be called “synthetic antiferromagnets (SAFs) structure.

Point 4 (round 2): In the scientific literature the " AFM domains " is a polynomial expression, which is associated with domain in antiferromagnet. Therefore, authors should use others words to indicate domains in the antiferromagnetically coupled ferromagnet layers.

Respond 4 (round 2): Your advice has been invaluable, and I have used “AFM-coupled domains” to replace “AFM domains”.

Point 8: lines 151-153: What means “but D = 0.1 and 0.25 mJ/m2”in the end of phrase “When the D varied between 0.1 mJ/m2 and 2.0 mJ/m2, the solid and dashed lines were overlapped for all the D values but D = 0.1 and 0.25 mJ/m2”?

Respond 8: The solid and dashed lines in Fig. 3a show the temporal central positions of the DWs in the upper and lower track, respectively. The lengths of the staggered domain region (Ls) for D = 0.1 and 0.25 mJ/m2 were as small as 5 and 10 nm (1 and 2 cells), respectively. And the length of the staggered domain region (Ls) is zero when D is bigger than 0.25 mJ/m2. So the solid and dashed lines were overlapped for all the D values but D = 0.1 and 0.25 mJ/m2.

Point 8 (round 2): may be authors would like say “but except D = 0.1 and 0.25 mJ/m2”.

Respond 8 (round 2): Thank you very much for your detailed suggestion. We have modified this part according to your suggestion. Line 159.

Point 11: The references 25 и 27 are indicated incorrectly.

Respond 11: Already modified the reference [27].

“[27] .Donahue MJ, Porter DJ (2002) OOMMF user s guide, version 1.2a3.”

Point 11 (round 2): The page in the reference {25] is not indicated.

Respond 11 (round 2): Already modified the reference [25].

“[25].Hrabec A , Porter N A , Wells A , et al. Measuring and tailoring the Dzyaloshinskii-Moriya interaction in perpendicularly magnetized thin films[J]. Physical Review B, 2014, 90(2):020402.”
